# The Relationship between Lifespan of Marine Bivalves and Their Fatty Acids of Mitochondria Lipids

**DOI:** 10.3390/biology12060837

**Published:** 2023-06-09

**Authors:** Aleksandra Anatolyevna Istomina, Avianna Fayazovna Zhukovskaya, Andrey Nikolaevich Mazeika, Ekaterina Andreevna Barsova, Victor Pavlovich Chelomin, Marina Alexandrovna Mazur, Olesya Alexandrovna Elovskaya, Andrey Alexandrovich Mazur, Nadezda Vladimirovna Dovzhenko, Yuliya Vladimirovna Fedorets, Alexander Alexandrovich Karpenko

**Affiliations:** 1Il’ichev Pacific Oceanological Institute, Far Eastern Branch, Russian Academy of Sciences, 690041 Vladivostok, Russia; s-istomina1@mail.ru (A.A.I.); avianna@poi.dvo.ru (A.F.Z.); chelomin@poi.dvo.ru (V.P.C.); olesya-sharova@mail.ru (O.A.E.); doreme_07@mail.ru (N.V.D.); zavertanova@poi.dvo.ru (Y.V.F.); 2Institute of the World Ocean, Far Eastern Federal University, 690922 Vladivostok, Russia; mazeyka.an@dvfu.ru (A.N.M.); barsova.ea@dvfu.ru (E.A.B.); 3A.V. Zhirmunsky National Scientific Center of Marine Biology, Far Eastern Branch, Russian Academy of Sciences, 690041 Vladivostok, Russia; proshinamarina94@gmail.com (M.A.M.); alex_karp@list.ru (A.A.K.)

**Keywords:** oxidative stress theory, peroxidation index, oxidative stress in vitro

## Abstract

**Simple Summary:**

Determining the physiological and biochemical causes of aging in animals is important both because of the potential health utility for humans and because aging is related to growth, reproduction processes, and the response of organisms to environmental conditions and stress. It is assumed that the peculiarities of the fatty acid composition of mitochondrial membranes (“membrane-pacemaker” theory of aging) can influence the rate of oxidative damage in cells, as well as the rate of the aging process. This property, in turn, may be fundamental for all living organisms. In this study, the fatty acid composition of gill tissues’ mitochondrial membranes, in marine bivalves, was determined and analyzed. The observed features in the lipid composition of mollusk mitochondria correlate well with the longevity of these animals.

**Abstract:**

Marine bivalves belonging to the *Mytilidae* and *Pectinidae* Families were used in this research. The specific objectives of this study were: to determine the Fatty Acids (FAs) of mitochondrial gill membranes in bivalves with different lifespans, belonging to the same family, and to calculate their peroxidation index; to compare the levels of ROS generation, malondialdehyde (MDA), and protein carbonyls in the mitochondria of gills, in vitro, during the initiation of free-radical oxation; to investigate whether the FAs of mitochondria gill membranes affect the degree of their oxidative damage and the maximum lifespan of species (MLS). The qualitative membrane lipid composition was uniform in the studied marine bivalves, regardless of their MLS. In terms of the quantitative content of individual FAs, the mitochondrial lipids differed significantly. It is shown that lipid matrix membranes of the mitochondria of long-lived species are less sensitive to in vitro-initiated peroxidation compared with the medium and short-lived species. The differences in MLS are related to the peculiarities of FAs of mitochondrial membrane lipids.

## 1. Introduction

The process of biological aging is characterized by a progressive decline in the efficiency of physiological functions. The ability to maintain the homeostasis of basic cellular processes weakens with age, which ultimately leads to an increased risk of many diseases and increases the probability of death [1].

Currently, the most widely accepted explanation of the mechanisms of aging is the free radical theory proposed by Harman in 1956. According to this theory, reactive oxygen species (ROS) formed during metabolism exhibit high reactivity and inevitably damage important biological structures (including lipids, proteins, and nucleic acids). The accumulation of damages is accompanied by a decrease in physiological functions, and it ultimately leads to the aging and death of the organism [2]. However, the status of this theory is far from certain, as some studies have found a lack of correlation between oxidative damage and lifespan, and the genetic manipulation of antioxidant pathways in invertebrate models (e.g., the nematode worm *Caenorhabditis elegans* and the fruit fly *Drosophila melanogaster)* have yielded variable results on life span, whereas studies in higher animals (e.g., the naked mole-rat *Heterocephalus glaber)* have not, generally, supported a role for oxidative stress in modulating longevity [3,4,5,6].

Later, it became evident that there is a close link between ROS generation (mainly in the respiratory chain of mitochondria) and the aging process [7,8]. Most of the oxygen consumed by the cell is involved in mitochondrial oxidative phosphorylation. During this process, a stepwise one-electron reduction in an oxygen molecule occurs, with the generation of its active forms (O_2_^•−^, OH^•^, H_2_O_2_) as intermediate products [9]. It has been shown that the rate of oxidative attacks, of ROS, on mitochondrial DNA is higher than on nuclear cell DNA [10]. At the same time, it turned out that the oxidative damage of mitochondrial DNA was inversely correlated with the lifespan of some birds and mammals since mutations, caused by damage in mitochondrial DNA, increased the aging process [10].

When explaining the mechanisms of aging, an attempt was made to combine the theory of oxidative stress (“free radical” theory) with the intensity of metabolism (“rate of living” theory). According to this viewpoint, organisms with a high metabolic rate are characterized by an increased production of oxyradicals that promote the rapid generation and accumulation of oxidative damage in the cell. On examples of certain representatives of mammals, birds, cephalopod mollusks, and the housefly (*Musca domestica*), by direct and indirect methods, it has been shown that ROS generation negatively correlates with lifespan [11,12,13].

However, as the experimental data were accumulated, this popular concept was considered to be erroneous. For example, it has been shown that spontaneous physical exercise and the associated increase in metabolic rate do not decrease the lifespans of mammals [1]. Another example is that birds and mammals have similar metabolic rates, but birds tend to live much longer than similarly sized mammals [14].

A relatively recent viewpoint has emerged, according to which the processes of aging and maximum lifespan (MLS) are closely connected to the susceptibility of a membrane lipid matrix to peroxidation, the key role in which is assigned to the nature of Fatty Acids (FAs). The composition of the FAs of membrane lipids subjected to oxidation correlates with the MLS of some birds and mammals, varies with body size [15,16], and is related to their MLS [17,18]. These studies played an important role in the development of “homeoviscous-longevity” theory and, later, the “membrane-pacemaker” theory of aging [19,20]. These theories assume that the characteristics of the FAs of mitochondrial membranes may influence the rate of oxidative damage in cells and the MLS of species.

It is known that the susceptibility of the same FAs to peroxidation increases exponentially with the number of double bonds of the carbon chain. Therefore, a single average value of susceptibility to peroxidation for any biological membrane, which has been named as peroxidation index (PI) [1,21], can be calculated from the profile of membrane FAs. The higher the value of the index, the more sensitive the lipid matrix is to oxidation.

The first indications of the relationship between the membrane composition and maximal lifespan were given by Pamplona and colleagues [22], who showed that the oxidation index PI of rat liver, guinea pig, and human mitochondria membranes correlated with their respective lifespan values. Later, it was shown that such a pattern is also typical for other tissues of humans and animals, including mammals, birds, and crustaceans [1].

Nevertheless, there are very few papers describing the applicability of this hypothesis to various invertebrate species. Despite the fact that there are extensive literature data on the FAs of lipids in invertebrate membranes, there are almost no papers linking it to the aging processes and lifespan of a particular species. For example, among invertebrates, the membrane lipid oxidation index has been calculated for individual representatives of bivalves [21,23]. The authors claim that there is a significant negative correlation between PI and the maximum lifespans (MLS) of these species. MLS is equivalent to the lifespan of the oldest observed specimen of a particular animal species, and it remains a frequently used trait in comparative biology [24].

Taking into account the fact that similar studies on phylogenetically similar species of marine mollusks have not been performed, we aimed to fill this gap and make some contribution to the development of this theory (“membrane-pacemaker” theory of aging). In addition, the specific objectives of this study were:-to determine the FAs of mitochondrial gill membranes, in bivalves with different lifespans belonging to the same family, and to calculate their peroxidation index;-to compare the levels of ROS generation, products of oxidative damage to lipids—malondialdehyde (MDA)—and protein carbonyls in the mitochondria of mollusk gills, in vitro, during the initiation of free-radical oxidation in the Fe-ascorbic acid model;-to investigate whether the FAs of gill mitochondrial membranes affect the degree of their oxidative damage and the MLS of species.

Marine bivalves belonging to the *Mytilidae* Families (*Mytilus trossulus* Gould, 1850, *Modiolus kurilensis* Bernard, 1983, *Crenomytilus grayanus* Dunker, 1853) and *Pectinidae* Families (*Chlamys farreri* Jones and Preston, 1904, *Swiftopecten swiftii* Bernardi, 1858, *Mizuhopecten yessoensis* Jay, 1857) were used in this research. Bivalves are genetically intermediate to classical invertebrate models of aging (e.g., worms and flies) and mammals. This provides a better opportunity to understand the evolution of stress-response pathways and organismic aging [7]. Recent studies have shown that bivalves are excellent models for aging research [25,26]. First, some individuals can reach a significant age: for example, *Arctica islandica* (507 years) or *Crenomytilus grayanus* (150 years). At the same time, among them, there are also short-lived species, such as surf clams (Family *Donacidae*), with species of no more than a 1 year lifespan, as well as *Mytilus trossulus* (6 years). Second, it is possible to study the different-aged species living in the same environmental conditions and, accordingly, experiencing similar fluctuations in the environmental temperature during the year. It is likely that such species should have an approximately constant FA composition of membranes. Third, some bivalves are capable of maintaining their metabolism, at a basic level, under stressful conditions in the shelf zone. Among the mechanisms for maintaining such a state, one is the low susceptibility to membrane lipid peroxidation. In addition, the composition of mollusk membranes is very different from that of endothermic animals. Plasmalogens and non-methylene-interrupted FAs are found in significant amounts in the membrane lipids of all molluskan organs. It is assumed that they significantly affect the liquid crystal structure of the lipid matrix and act as retarders of the peroxidation processes in the membrane. Their presence also increases the antioxidant activity of lipids [8,27]. The variations in membrane FA composition may be an important missing link in the problem of explaining aging and the mechanisms that determine the maximum lifespan specific to each species. This is a testable hypothesis that requires further experiments.

## 2. Materials and Methods

### 2.1. Site of Bivalves Collection and Material

Mature mollusks were collected during the post-spawning period, in November 2021, in the waters of the Alekseev Bay and Stark Strait in the Sea of Japan (Figure 1).

The biological characteristics of bivalves are shown in Table 1.

Mollusks were transported to the aquarium of the A.V. Zhirmunsky National Scientific Center of Marine Biology, where they were maintained at a constant temperature of 16 °C for 3 days to relieve the stress of transportation.

For FA analysis, 2 g of gills were obtained from one individual, for a total of 5 individuals for each species; for *M. trossulus*, tissue from 16 individuals was pooled for a total of 80 individuals. The mitochondria obtained for each mollusk species were separated into three samples (*n* = 3). For ROS, MDA, and carbonyl analysis, mitochondria were obtained from gills weighing 0.6 g. For most mollusks, 1 sample was 1 individual, for a total of 6–8 samples (*n* = 6–8). For *M. trossulus,* 1 sample was an assemblage of 3 individuals, for a total of 6–8 samples (*n* = 6–8). The isolated gills were frozen in liquid nitrogen and stored for not more than 1 month before analysis. All procedures in the present work, as well as the mollusk disposal methods, were approved by the Commission on Bioethics at the V.I. Il’ichev Pacific Oceanological Institute, Far Eastern Branch of Russian Academy of Science (protocol №16 and date of approval 15 April 2021), Vladivostok, Russia.

The individual age of scallops and *M. trossulus* was estimated by growth retardation rings on the surface of the shell. The data were comparable with the growth curves obtained for these species by other authors (Table 1). The age of *C. grayanus* and *M. kurilensis* was determined by the curves of group linear growth (Table 1).

### 2.2. Mitochondria Isolation

Gills were homogenized on ice (1:5, weight/volume). Mitochondria were isolated in 0.5 M NaCl in a 0.05 M Tris-HCl (pH 7.5) medium containing 0.25 M sucrose, 1 mM EDTA, and 0.1 mM PMSF. The medium for homogenization was pre-blown with argon. The homogenate was centrifuged at 1000× *g* for 12 min to remove large residual cells and nuclei. The resulting supernatant was centrifuged at 12,000× *g* for 30 min. The mitochondria were washed from sucrose 3 times in 0.5 M NaCl in 0.05 M Tris-HCl (pH 7.5).

### 2.3. Biochemical Analysis

ROS levels were determined by the oxidation of DHR 123 (dihydrorhodamine 123) to fluorescent rhodamine 123 [35]. MDA content was determined by a color reaction with 2-thiobarbituric acid [36]. Protein carbonyl groups were determined by the alkaline method [37], and protein concentration was determined by the modified Lowry method [38].

### 2.4. Oxidative Stress In Vitro

The oxidative stress reaction was triggered by adding Fe^2+^ and ascorbic acid (50 μM and 100 μM in the incubation medium, respectively) to mitochondria at 20 °C for one hour for MDA and carbonyl determination, as well as 15 min for ROS determination.

### 2.5. Determination of FAs

Lipid FAs were analyzed in the form of methyl esters using an Agilent 3700 chromatograph with a flame ionization detector. We used a Carbowax-20 M capillary column 25 m × 0.2 mm, a helium carrier gas, and a thermostat temperature of 200 °C [39]. FAs were identified by comparing the relative retention times of their methyl esters with the FA methyl esters of the standard mixture and the “carbon numbers” values [40]. The percentage of acids was calculated according to the method of Carrol [41]. FA methyl esters were obtained according to the method of Carreau and Dubacq [42].

### 2.6. Statistical Analysis

Statistical processing of the results was performed using Statistica 7. Breakdown and one-way ANOVA, as well as Statistics by Groups, Post-hoc were used to assess the reliability of parameter changes. Significance was established at *p* < 0.05.

## 3. Results

### 3.1. FAs in Mitochondrial Membranes of Mollusk Gill Cells

According to the results of the analysis presented in Table 2, the qualitative composition of the FAs of gill cell mitochondria lipids is uniform in all the studied marine bivalves, regardless of their MLS. However, in terms of the quantitative content of individual FAs, the mitochondrial lipids of mollusks differed significantly.

Despite significant variations (from 21.06 to 47.8%) in the content of total saturated fatty acids (SFAs), in all representatives of the *Mytilidae* Families and the *Pectinidae* Families, the palmitic and stearic acids (16:0 and 18:0) dominated. At the same time, the lowest content of SFAs was found in the mitochondrial lipids of the scallop *S. swiftii*, and the maximum was in the Pacific mussel *M. trossulus*. In general, it turned out that the SFAs in mitochondrial lipids in short-lived species was higher than in medium-lived and long-lived bivalves (Table 2).

In most mollusks, oleic acid (18:1 *n−*7) dominated among monounsaturated fatty acids (MUFAs), except for *C. grayanus* and *M. yessoensis*, in which eicosenoic acid (20:1 *n−*9) MUFAs predominated. The total content (MUFAs) in lipids also varied widely (from 11.4 to 20.5%) in the mollusks studied: the minimum amount was observed in the mitochondria of *M. trossulus*, and the maximum was in *C. grayanus* and *C. farreri*.

Non-methylene-interrupted fatty acids (NMI FAs), represented mainly by docosadienoic acid (22:2), were found in the FA composition of the mitochondria of marine mollusks. The greatest variation in the content of this acid was observed in pectinids: from 1% in the scallop *C. farreri* to 11.7% in the scallop *M. yessoensis*. The 22:2 level in representatives of the *Mytilidae* Family increased in the series: *M. trossulus*—*M. kurilensis*—*C. grayanus*; in representatives of the *Pectinidae* Family—in the series*: C. farreri*—*S. swiftii*—*M. yessoensis*. Regarding NMI FAs, both by the content of individual 22:2 and by the total level of NMI FAs, the total content of which varies from 2.76% (*C. farreri*) to 13.7% (*C. grayanus*), a direct connection with MLS is observed in species of the relevant family.

In representatives of the *Pectinidae* Family, polyunsaturated fatty acids (PUFAs) dominated in the FAs of gill mitochondrial cell membranes; their amount was greater than the total sum of SFAs and MUFAs. The *Mytilidae* Family showed a different pattern: the level of PUFAs did not exceed, and in some cases, it was lower than the total sum of saturated and monounsaturated acids (Table 2). Docosahexaenoic acid (22:6 *n−*3) significantly prevailed in the PUFA of mitochondria lipids, especially in pectinids. Among PUFAs, the ratio of *n−*3/*n−*6 acids varied from 2.10 to 4.92, with the minimum values observed in the *C. grayanus* and the *M. yessoensis*, and the maximum values were characteristic of the Pacific mussel *M. trossulus* and the Zhikong scallop *C. farreri* (Table 2).

### 3.2. PI of Mitochondrial Membranes

On the basis of the composition of FAs, according to the formula given in [21], the lipid peroxidation index was calculated, the values of which are shown in Table 2. From the analysis of these values, it follows that the propensity to oxidation of the FA lipids of mitochondrial membranes, in representatives of the *Pectinidae* Family, is higher than that in representatives of *Mytilidae* Family. At the same time, no correlation between the obtained values of PI and MLS of bivalves was revealed.

### 3.3. Constitutive Levels of ROS, MDA and Carbonyls

The basal levels of ROS generation and MDA content in mitochondria were highest in the long-lived Gray’s mussel *C. grayanus* and the coastal scallop *M. yessoensis*, as compared with the medium and short-lived representatives of the respective families (Figure 2). In general, representatives of the *Mytilidae* Family differed from those of the *Pectinidae* Family (*S. swiftii, C. farreri*) in higher MDA content in the mitochondria of gill cells. At the same time, *C. grayanus* and *M. kurilensis* had a lower level of ROS generation compared to *M. yessoensis* and *C. farreri*, respectively. No interspecific differences in the content of protein carbonyls were found in any of the bivalves studied.

### 3.4. Induction of Oxidative Stress In Vitro

The results of this series of experiments showed that, when free-radical processes were initiated using the Fenton reaction, the lowest level of oxygen radical generation was registered in the mitochondria of the long-lived mussel Gray’s *C. grayanus* and the scallop *M. yessoensis* in contrast to the short-lived mussel *M. trossulus* and the scallop *C. farreri* (Figure 3). A similar pattern was observed in the formation of the main product of lipid oxidation—MDA.

Under these conditions of ROS generation initiation, the least amount of MDA accumulated in the mitochondrial lipids of long-lived mollusks (*C. grayanus* mussel and *M. yessoensis* scallop) compared with short-lived ones (*M. trossulus* and *C. farreri*). Mitochondrial membranes did not differ in protein carbonyl levels in bivalves from both families (Figure 3).

## 4. Discussion

### 4.1. Specific Features of FAs in Gill Mitochondria Lipids

Unsaturated fatty acids are easily subjected to oxidative damage in the cell, and the rate of oxidation increases with the number of double bonds. Therefore, unlike SFAs and MUFAs, which are relatively resistant to oxidation, PUFAs are easily and rapidly oxidized [8,43]. According to *homeoviscous theory*, the liquid crystalline state of the lipid matrix, necessary for the function of biological membranes, is maintained by regulating the degree of unsaturation of the acyl chains of phospholipids. In this respect, using mammalian and avian representatives as an example, it has been shown that the high unsaturation of membrane lipid FAs is associated with an increased level of oxidative lipid damage, but it negatively correlates with MLS [17,44]. Thus, in representatives of long-lived mammals, as compared to species with shorter lifespans (short-lived ones), a decrease in the ratio of acids with 4 or 6 double bonds and an increase in the level of FAs with 2 and 3 double bonds were found. At the same time, as noted by the authors, not only was a significant increase in lipid resistance to peroxidation observed but the corresponding fluidity of the lipid matrix was also maintained, and all the most important functions of membranes (receptor, ion transport, metabolite transport, etc.) were performed [8,44].

In fact, the results of this study of the FAs of mitochondrial lipids, in representatives of two families of marine bivalves, confirm this theory.

On the basis of the SFAs/PUFAs ratio, the authors showed that the amount of SFAs in gill membrane lipids was higher in representatives of the *Mytilidae* Family compared with the *Pectinidae* Family. It turned out that the long-lived *C. grayanus* and *M. yessoensis* had a lower SFAs/PUFAs ratio compared to the short-lived *M. trossulus* and *C. farreri* from the respective families. In addition, as in mammalian representatives, the proportion of FAs with 2 and 3 double bonds, in relation to FAs with 4 and 6 double bonds, was higher in long-lived species vs. short-lived species (Table 2).

It is known that the tendency of oxidation of acyl chains of lipids is determined not only by the degree of unsaturation but, also, by the position of double bonds. It was found that *n−*3 PUFAs are oxidized faster than *n−*6 PUFAs. Accordingly, membranes enriched with phospholipids with *n−*6 FAs are more stable in response to unfavorable environmental factors [45]. In addition, the ratio of these PUFAs (Σ*n−*3/Σ*n−*6) is an index characterizing the viscosity/liquidity of the lipid matrix of biological membranes. The lower is the ratio of *n−*3/*n−*6, the lower is the viscosity of the lipid matrix, but the higher is the resistance of lipids to oxidation, which is beneficial for the stability of membrane processes. In this respect, the paper of Valencak and Ruf [46] should be particularly noted. The authors revealed a negative correlation between the increase in the *n−*3/*n−*6 ratio in the skeletal muscle lipids of mammalian representatives and their lifespan. This interesting tendency is also clearly seen in marine bivalves. In this study, it was found that, in the mitochondrial lipids of long-lived mussel *C. grayanus* and scallop *M. yessoensis*, the ratio of Σ*n−*3/Σ*n−*6 is lower in comparison with the medium-lived and short-lived representatives of the respective families (Table 2). Based on this, there is every reason to believe that the lipid matrix of mitochondrial membranes of long-lived species is more stable in response to the effects of unfavorable environmental factors.

In addition to the above characteristics of mitochondrial membrane lipids, the presence of NMI FAs draws attention, which can also have a significant influence on the structure and function of biological membranes. These unusual FAs can act as “structural antioxidants”, slowing down the lipid matrix peroxidation processes [47]. The obtained results showed that the acids [Σ20:2 (5,11); 20:2 (5,13); 22:2] were present in far greater amounts in the lipids of long-lived *C. grayanus* and *M. yessoensis* mollusks than in the lipids of medium-lived and short-lived species from the respective families (Table 2). Therefore, it is logical to assume that the lifespans of the studied mollusks are related to the presence of these FAs in the lipids, which protect mitochondrial membranes from oxidative damage to a certain extent.

In general, the observed features in the lipid composition of mollusk mitochondria, through the presumed effect on lipid matrix oxidability, correlate well with the lifespans of these animals. Although the integral index (PI), calculated based on FA composition, demonstrated an increased sensitivity to the oxidative degradation of lipid membranes of representatives of the *Pectinidae* Family vs. representatives of the *Mytilidae* Family, it showed no relationship with the lifespans of bivalves. In this respect, the results of these studies and reasoning are consistent with those of Valencak and Ruf [46], who also found no correlation between the skeletal muscle lipid oxidation index (PI) and lifespan in 42 mammalian species.

The absence of such correlation calls into question the correctness of the calculation of this index, which does not consider additional factors influencing lipid oxidability. Among them, it should be emphasized that the high content of etheric lipids with alkyl and alkenyl fat radicals is characteristic of bivalves, whose contribution to lipid matrix oxidation of membranes is practically unstudied [48].

### 4.2. Constitutive Levels of ROS, MDA and Carbonyls in Gill Mitochondria

The “membrane-pacemaker” theory of aging suggests that lifespan can be related not only to lipid matrix oxidizability (based on PI) but, also, to the rate generation of ROS in the cell. The main source of ROS generation in the cell is the electron-transport chain localized in mitochondrial membranes. Taking into account their high reactivity, these ROS can initiate free-radical processes and cause the destruction of membrane lipids, proteins, and damage to mitochondrial DNA. In the latter case, there is strong evidence that the rate of aging is closely related to the frequency of mutations occurring in mitochondrial DNA [10]. In the lipid matrix of membranes, unsaturated fatty acids, especially PUFAs, become the preferred target for ROS. After the initiation of free-radical processes through a cascade of reactions, these FAs decompose to form highly reactive carbonyl compounds, such as malondialdehyde (MDA) and 4- hydroxynonenal (4-HN), which exhibit various cytotoxic and genotoxic properties [49]. It is likely that, through the regulation of ROS generation, mitochondria play a key role in preventing the formation and accumulation of various destructive damages affecting aging processes.

This opinion is based on the results of mammalian and bird studies in which it has been shown that, regardless of oxygen uptake rate, long-lived species show low rates of mitochondrial radical generation and contain lower constitutive levels of antioxidant activity [7,8,12,14]. Nevertheless, it has been shown that, in the long-lived (naked mole rat) *Heterocephalus glaber*, the endothelial and smooth muscle cells of carotid arteries and aorta produce comparable—or even higher—levels of ROS compared to short-lived mice [50].

The results of these studies also do not fit the general hypothesis. In long-lived *C. grayanus* and *M. yesonensis*, a relatively high baseline level of ROS generation and elevated MDA content in mitochondria were observed compared with other representatives of their families. Moreover, all this is realized against the background of a relatively low baseline level of antioxidant potential, including the activity of antioxidant enzymes and low molecular weight antioxidants [51].

Previously, in comparative studies of bivalves belonging to different families, it was shown that the isolated gill and heart mitochondria of long-lived *Arctica islandica* generated less ROS compared to short-lived *Mya arenaria*, *Spisula solidissima,* and *Mercenaria mercenaria* [27,51]. At the same time, the short-lived scallop *Argopecten irradians* and the long-lived *Tridacna derasa* did not significantly differ in ROS generation in gills, adductor muscles, and heart cells [52]. There was also no difference in the carbonyl content of the gills and adductor muscle in these species.

Comparing these results with the above examples, it is logical to assume that, in long-lived mollusks, against the background of low antioxidant protection, the hydrophobic component probably plays an important role in the mechanisms maintaining the oxidative stability of the lipid membrane matrix. These ideas, to a certain extent, were confirmed in the authors’ experiments with the induction of mitochondrial lipid peroxidation initiated by the Fenton reaction. This approach makes it possible to not only estimate the potential ability of mitochondria to generate ROS but, also, to reveal the integral vulnerability of the hydrophobic matrix to oxidative degradation in case of oxidative stress.

### 4.3. Response to Induced Oxidative Stress In Vitro

The results showed that the mitochondria of long-lived *C. grayanus* and *M. yessoensis* produced lower levels of ROS and less MDA as compared to medium and short-lived representatives of the respective families, indicating greater resistance of their lipid matrix to in vitro-induced oxidative damage. As far as one can estimate from the published data, the presented results are not only characteristic for the study species. It was also found that exposure of the scallop *Argopecten irradians* to paraquat, rotenone, or organic hydroperoxide causing oxidative damage of mitochondria was accompanied by a faster death of these short-lived mollusks compared to the long-lived *Mercenaria mercenaria, Arctica islandica,* and *Tridacna derasa* [53,54]. Moreover, there is evidence that experiments *in vitro*, fibroblasts, and lymphocytes of long-lived vertebrate species show increased resistance to induced oxidative stress [55].

When analyzing the results of mollusk mitochondrial resistance to the in vitro-induced oxidative damage of membrane lipids, the authors found two opposite trends. In representatives of the *Mytilidae* Family, in response to Fenton’s reagents, mitochondria generated high levels of ROS, which nevertheless led to an insignificant accumulation of MDA. In similar experiments on representatives of the *Pectinidae* Family, a different picture was observed: against the background of an insignificant level of ROS generation, we detected a significant increase in the MDA content of mitochondrial membrane lipids. Although it is beyond the scope of this study to investigate the reasons for these peculiarities, the authors should admit that the mitochondrial membranes of these two families differ significantly not only in lipid matrix accessibility to peroxidation but, also, in the function of ROS generation centers.

As the experimental data accumulate, it becomes more and more evident that different mechanisms of stabilization of not only the lipid matrix but proteins can make a certain contribution to the processes ensuring cell resistance to stress and lifespan [51,56]. Nevertheless, according to the experimental data, the authors found no changes in the content of protein carbonyls at neither the baseline nor after ROS generation in bivalves with different lifespans. Apparently, the reparation processes of the damaged mitochondrial membrane proteins, with the participation of proteosomal and autophagic mechanisms, are stable and exhibit resistance to short-term exposure to oxidative stress in these representatives of marine mollusks. Regarding the poor study of this issue, the authors consider it necessary to perform further studies to identify the mechanisms maintaining the stability and integrity of membrane protein components with the involvement of representatives of other taxonomic groups with different lifespans.

## 5. Conclusions

The common features of the relationship between the FA composition of gill mitochondrial membranes and the MLS of species are revealed only in a comparative analysis of mollusks having a common origin within a family. The response to in vitro-induced oxidative stress also has a relationship with the MLS of species belonging to the same family.

The basal levels of ROS and MDA formation in gill mitochondria are higher in the long-lived *C. grayanus* and *M. yessoensis* vs. medium and short-lived representatives of the respective families, and the gill mitochondrial membranes of these species are more resistant to *in vitro*-induced oxidative stress (low levels of ROS and MDA).

It is likely that an important mechanism of lifespan maintenance in *C. grayanus* and *M. yessoensis* is a specific FA composition of mitochondrial membranes. It is characterized by a lower ratio of SFAs/PUFAs and *n−*3/*n−*6, a higher ratio of the sum of FAs with 2 and 3 bonds and the sum of FAs with 4 and 6 bonds, and higher content of the sum of NMI FAs vs. medium and short-lived species of the respective family.

## Figures and Tables

**Figure 1 biology-12-00837-f001:**
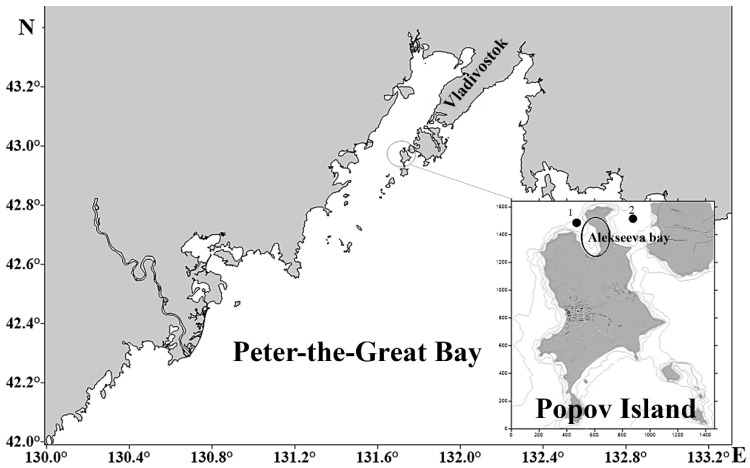
The location of the sampling sites in the Peter-the-Great Bay of the Japan Sea (Russia) (1—the collection site of mussels, 2—scallops).

**Figure 2 biology-12-00837-f002:**
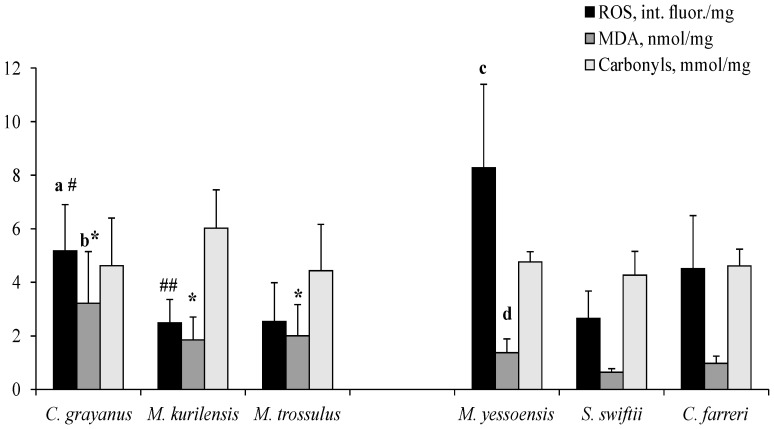
Constitutive levels of ROS (reactive oxygen species), MDA (malondialdehyde), and carbonyls in gill mitochondria. Significance of differences between: a—*C. grayanus* vs. *M. kurilensis* and *M. trossulus*; b—*C. grayanus* vs. *M. kurilensis*; c—*M. yessoensis* vs. *S. swiftii* and *C. farreri*; d—*M. yessoensis* vs. *S. swiftii* and *C. farreri*; **#—***C. grayanus* vs. *M. yessoensis*; * *C. grayanus, M. kurilensis* and *M. trossulus* vs. *S. swiftii* and *C. farreri*; **##—***M. kurilensis* vs. *C. farreri* (*n* = 6–8; Post-hoc, *p* < 0.05).

**Figure 3 biology-12-00837-f003:**
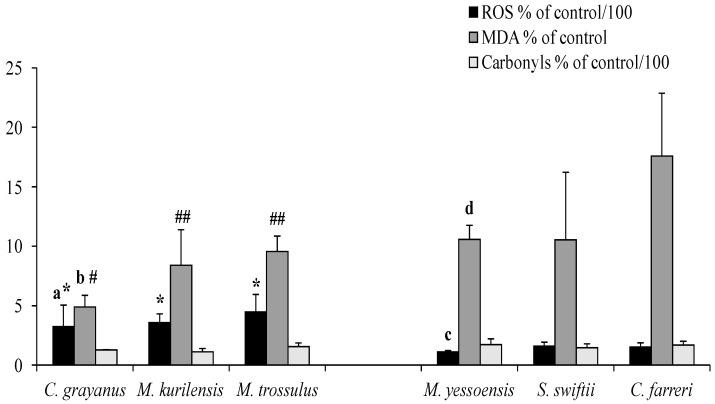
Response to induced oxidative stress in vitro. Significance of differences between: a—*C. grayanus* vs. *M. trossulus*; b*—C. grayanus* vs. *M. kurilensis* and *M. trossulus;* c—*M. yessoensis* vs. *S. swiftii* and *C. farreri*; d—*M. yessoensis* vs. *C. farreri*; *—*C. grayanus, M. kurilensis* and *M. trossulus* vs. *M. yessoensis, S. swiftii* and *C. farreri*; **#—***C. grayanus* vs. *M. yessoensis, S. swiftii* and *C. farreri*; **##—***M. kurilensis* and *M. trossulus* vs. *C. farreri* (*n* = 6–8; Post-hoc, *p* < 0.05).

**Table 1 biology-12-00837-t001:** Biological characteristics of bivalves.

Species	Length, mm	Approximate Age, Years	MLS, Years	Reference
*Mytilidae*				
*Crenomytilus grayanus*	116.9 ± 5.0	24	150	[28,29]
*Modiolus kurilensis*	111.7 ± 4.9	20	61	[29,30]
*Mytilus trossulus*	42.2 ± 3.9	4	6	[31]
*Pectinidae*				
*Mizuhopecten yessoensis*	132.0 ± 11.2	5	22	[32]
*Swiftopecten swiftii*	83.9 ± 5.2	4	15	[29,33]
*Chlamys farreri*	92.3 ± 5.3	4	9	[34]

Note: MLS—maximum lifespan.

**Table 2 biology-12-00837-t002:** Fatty acids (%) from gill mitochondria lipids. Values are mean ± SD, *n* = 3.

Fatty Acid	*Mytilidae*			*Pectinidae*		
	*C. grayanus*	*M. kurilensis*	*M. trossulus*	*M. yessoensis*	*S. swiftii*	*C. farreri*
12:0	1.0 ± 0.0	0.7 ± 0.2	1.1 ± 0.1	0.7 ± 0.0	0.7 ± 0.2	1.0 ± 0.7
14:0 ai	0.2 ± 0.0	0.7 ± 0.1	0.8 ± 0.0	0.5 ± 0.0	0.3 ± 0.1	0.8 ± 0.2
14:0	0.6 ± 0.0	0.3 ± 0.1	0.7 ± 0.0	0.1 ± 0.0	0.4 ± 0.1	0.1 ± 0.0
15:1 n−7	1.0 ± 0.0	1.8 ± 0.0	1.6 ± 0.3	0.4 ± 0.0	0.4 ± 0.1	0.6 ± 0.0
16:0	17.9 ± 0.9	15.9 ± 0.3	19.0 ± 0.1	12.5 ± 0.6	12.7 ± 0.8	13.7 ± 0.7
16:1 *n−*9	2.1 ± 0.1	2.0 ± 0.5	2.7 ± 0.3	1.3 ± 0.0	-	2.2 ± 0.2
16:1 *n−*7	2.3 ± 0.1	2.3 ± 0.1	1.9 ± 0.0	1.4 ± 0.0	2.3 ± 0.5	1.2 ± 0.0
17:0 i	0.6 ± 0.0	0.5 ± 0.0	0.5 ± 0.0	0.3 ± 0.0	0.9 ± 0.1	0.1 ± 0.0
17:0 ai	1.1 ± 0.1	1.8 ± 0.1	1.8 ± 0.0	1.2 ± 0.0	0.3 ± 0.1	2.3 ± 0.0
17:0	0.8 ± 0.0	1.9 ± 0.1	1.7 ± 0.0	1.0 ± 0.0	0.9 ± 0.2	-
18:0 i	2.8 ± 0.1	2.5 ± 0.2	4.0 ± 0.1	0.9 ± 0.0	1.2 ± 0.3	1.3 ± 0.1
18:0	9.6 ± 0.5	10.9 ± 0.3	19.8 ± 3.1	4.8 ± 0.2	3.3 ± 0.6	9.7 ± 0.1
18:1 *n−*9	0.7 ± 0.0	-	-	-	0.1 ± 0.1	1.0 ± 0.0
18:1 *n−*7	4.3 ± 0.2	5.1 ± 0.3	2.3 ± 1.6	4.3 ± 0.2	4.5 ± 0.4	5.4 ± 0.4
18:2 *n−*6	1.0 ± 0.0	2.4 ± 0.5	1.6 ± 0.3	2.0 ± 0.1	2.2 ± 0.2	2.5 ± 0.1
18:2 *n−*4	0.4 ± 0.0	1.5 ± 1.5	0.5 ± 0.2	0.6 ± 0.0	0.1 ± 0.0	0.3 ± 0.1
18:3 *n−*6	-	-	0.2 ± 0.0	0.2 ± 0.0	0.2 ± 0.0	-
18:3 *n−*3	0.8 ± 0.0	1.2 ± 0.0	0.3 ± 0.0	0.3 ± 0.0	0.2 ± 0.0	0.1 ± 0.0
20:0-i	1.8 ± 0.1	1.7 ± 0.1	0.4 ± 0.1	-	-	0.8 ± 0.1
18:4 *n−*3	0.2 ± 0.0	-	0.4 ± 0.0	-	0.1 ± 0.0	-
20:1 *n−*13	2.2 ± 0.1	0.6 ± 0.1	0.8 ± 0.2	2.0 ± 0.1	1.8 ± 0.0	2.9 ± 0.1
20:1 *n−*9	5.8 ± 0.3	3.4 ± 0.1	1.9 ± 0.3	5.5 ± 0.3	4.2 ± 0.5	4.2 ± 0.2
20:1 *n−*7	2.2 ± 0.1	3.9 ± 0.0	1.3 ± 0.2	1.0 ± 0.1	0.6 ± 0.3	2.5± 0.1
20:2 (5,11)	3.9 ± 0.2	1.9 ± 0.0	1.8 ± 0.1	5.4 ± 0.3	5.5 ± 0.6	0.9 ± 0.0
20:2 (5,13)	1.6 ± 0.1	0.6 ± 0.1	1.1 ± 0.2	1.7 ± 0.1	1.0 ± 0.1	0.7 ± 0.1
20:4 *n−*6	5.5 ± 0.3	5.7 ± 0.5	3.5 ± 0.3	6.7 ± 0.3	5.8 ± 1.2	3.9 ± 0.2
20:5 *n−*3	4.2 ± 0.2	6.2 ± 0.5	6.0 ± 0.4	3.9 ± 0.2	4.7 ± 0.6	9.7 ± 0.3
22:2	8.2 ± 0.4	5.2 ± 0.1	4.1 ± 0.2	11.7 ± 0.6	9.0 ± 1.2	1.0 ± 0.0
22:6 *n−*3	8.6 ± 0.4	9.7 ± 0.3	9.3 ± 0.4	22.1 ± 1.1	25.8 ± 1.4	21.7 ± 1.3
Total	91.1 ± 4.6	90.4 ± 0.9	87.4 ± 4.0	93.6 ± 4.7	89.5 ± 7.2	92.0 ± 1.3
SFAs	36.3 ± 1.8	37.0 ± 0.0	47.8 ± 0.5	22.36 ± 1.1	21.0 ± 1.9	30.3 ± 1.4
MUFAs	20.5 ± 1.0	19.0 ± 1.0	11.4 ± 3.1	16.12 ± 0.8	13.4 ± 0.4	20.4 ± 0.6
PUFAs	34.3 ± 1.7	34.4 ± 0.1	28.2 ± 1.4	55.13 ± 2.8	55.0 ± 4.9	43.6 ± 5.7
∑*n−*3	13.6 ± 0.7	17.1 ± 0.8	15.4 ± 1.0	26.55 ± 1.3	30.8 ± 1.9	31.6 ± 1.6
∑*n−*6	6.5 ± 0.3	8.1 ± 1.0	5.2 ± 0.4	9.03 ± 0.5	8.3 ± 1.5	6.4 ± 0.4
*n−*3/*n−*6	2.10	2.11	2.98	2.94	3.69	4.92
SFAs/ PUFAs	1.05	1.08	1.70	0.41	0.38	0.70
∑NMI FAs	13.7 ± 0.01	7.75 ± 0.24	6.93 ± 0.15	18.9 ± 0.1	15.68 ± 1.63	2.76 ± 0.22
∑2*n*, 3*n*/ ∑4*n*, 6*n*	1.11	0.84	0.73	0.77	0.58	0.22
PI	124.5 ± 6.2	146.7 ± 6.6	129.7 ± 7.9	238.2 ± 11.2	267.3 ± 19.5	252.7 ± 13.6
MLS	150	61	6	22	15	9

Note: MLS—maximum lifespan; PI—peroxidation index; SFAs—saturated fatty acids; MUFAs—monounsaturated fatty acids; PUFAs—polyunsaturated fatty acids; NMI FAs—non-methylene-interrupted fatty acids.

## Data Availability

Not applicable.

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
