# Peer review of "The Relationship between Lifespan of Marine Bivalves and Their Fatty Acids of Mitochondria Lipids"

_biology, 2023, doi:10.3390/biology12060837_

Round 1

Reviewer 1 Report

I do not have many objections to the submitted manuscript. Nevertheless, I would suggest developing two points:

1. Introduction. It is worth citing articles criticizing the validity of free radical theory and showed limitations and concerns regarding it.

2. I cannot agree with the following statement of the authors "Adult mollusks were selected by size so that they were in a similar physiological state, approximately in the middle of their MLS". In fact, the selection of organisms by age was not very "successful" (actual age vs. MLS). E.g. Mytilus trossulus was at an age corresponding to 75% of it's MLS, and Crenomytilus grayanus about 15% of MLS. This may have affected the results obtained and their comparability. This fact should be clearly indicated in the text and addressed. 

Author Response

Thank you for your comments. The following is my reply to your comments.

Reviewer 2 Report

Comments to the Author

Title:

Some Aspects of the Relationship between Lifespan of Marine 2 Bivalves and their Fatty Acids of Mitochondria Lipids

Biology-2404044
The manuscript deals with the marine mollusk life span determination through peroxinectin index, free radicals’ content and ROS regeneration and
qualitative membrane lipid composition is 21 uniforms in all the studied marine bivalves, regardless of their MLS. Biological aging of the marine animals through this approach is appreciable, mitochondrial isolation, ROS level assessment and biochemicals assays were carried out to investigate this study , though I have few clarifications

However the present form of manuscript needs compliance of certain queries and suggested modification before it is recommended for publication:

 1.   In the introduction, it is unclear why this species was chosen in this study. Biological aging has to be elaborated; significance of the study needs to incorporated. There is insufficient information on the current status and problems of marine organisms, or I don't see any ecological of economic importance in background of this species in the introduction.   

2.      How the FA composition helps to understand  the aging is not clear?

3.      In the discussion, there is a lack of comparative discussion with the previous studies on other gastropod  characterization

4.      In the results, the micrographs (Fig. 3, 4) showing different types of expression patterns do not meet the standard of the journal. The pictures are not refined and have low magnification to identify the expression level features and immune functions.
5.      In the discussion, there is a lack of comparative discussion with the previous studies on gastropod hemocyte characterization. Although this study was conducted on mollusc, the author mainly focused to identify the age of the mollucs. In gastropods, the presence of mitochondria and other FA not always confirmed. Such discrepancies among different
gastropods should be discussed.

6. In its current state, the level of English throughout your manuscript does not meet the journal's desired standard. Please check the manuscript and refine the language carefully.

7. The spelling for organisms and species  should be the same.

8.  Completely identify genus and species of organism the first time they are listed and check for spelling.

9.  The introduction and results sections should be revised for common punctuation and run-on sentences.

10. Make sure that acronyms or words are spelled the same throughout the paper.

11. Details of  immunological assays need to be discussed in elaborative way and  along with these information if possible authors can include  image of experimental animals and microscopic images for concurrent evidences to support this study. Still this study can be recommend for major revision.

NIL

Author Response

(The authors gave the same response as above.)

Reviewer 3 Report

The manuscript “Some aspects of the relationship between lifespan of marine bivalves and their fatty acids of mitochondrial lipids” determined the fatty acids of mitochondria in gills of bivalve species with different lifespans, calculated the peroxidation index and compared the basal and induced oxidation levels. Even though this is an interesting topic and a study with many potentials, the data represented in the current form is limited, the results are mostly descriptive, and the conclusion is arbitrary to test so many things and theories. Therefore, I recommend rejection but encourage resubmission with major revision and thorough improvement.

Here are the major points the authors may consider to improve their manuscript:

1)   The English is obscure and very difficult to follow, with many long sentences and grammatic errors. For example, at line 58, “as the experimental data were accumulated” should be “as the experimental data accumulated”. There are may other examples that were not shown here, but please carefully check the language.

2)   How to define the bivalve species used in this study as long-lived or short-lived species? There is only limited information from table 1, for example, the MLS of 3 species from Pectinidae were 22, 15, 9 respectively, then the authors defined 22 years M. yessoensis as the long-lived? Are there other species from Pectinidae living longer than 22 years? Which then may lead M. yessoensis medium-lived or short-lived?

3)  The authors determined the age of the bivalve samples according to their body size? Is this method precise or qualified as the standard method for bivalve age determination?

4)  At line 107, the authors mentioned that the bivalve species used in this study have similar body size, but from table 1, the M. trossulus is obvious smaller than other species, which has also made the gill sample strategy different from other species in the methods section. Therefore, is this species suitable for this study?

5)  For Chlamys farreri, the authors used many time as Ch. farreri at line 182, 190, 200 and so on, why not use the latin name as C. farreri ? Moreover, at line 200, “Japanese scallop Ch. farreri”. The common name for C. farreri is “Zhikong scallop” or “Chinese scallop”, and as a matter of fact, M. yessoensis is more commonly used as “Japanese scallop”.

6)  There are only 2 tables and 2 figures in this study, with table 1 and figure 1 for the sample information, and only table 2 and figure 2 for the results. Therefore, the supporting result is limited in this study, which is not solid to support the conclusion, and the conclusion is too arbitrary.

7)  In figure 2, the significant difference labels “a, b, c, a*, b*” is confusing, please clarify. Moreover, species latin names should be in italic, such as at line 231-233. Moreover, at line 65 “FAs”, the first-time appearance in the manuscript should be used the full name “fatty acids (FAs)”.

8)  In the discussion, there is only 4.1 with out 4.2, 4.3 ….. Then it is not necessary to have the 4.1 anymore.

9)  In the introduction and discussion, there are too may paragraphs even with only 3-4 lines. This indicates the logic of this manuscript is not clear, that is why the manuscript is very obscure and difficult to follow. The authors need to make the story clearer and more convincing in a more logic way.

10)  Again, the title “some aspects of …..” also indicates that the logic and conclusion of this study lack of focus and try to use limited information to test too many things and theories.

The English is obscure and very difficult to follow, with many long sentences and grammatic errors.

Author Response

(The authors gave the same response as above.)

Round 2

Reviewer 2 Report

This paper can be accepted all the comments mentioned  was rectified. Hence I recommend for publication

Author Response

Спасибо за ваши Коментарии. Далее мой ответ на ваши комментарии.

Reviewer 3 Report

   This is the most perfunctory “response to reviewer” that I have ever met. The authors just used “It is corrected. to response to most of the comments, which in fact either I can not find where it is corrected or they actually did not correct anything.

   For example, the authors replaced the figure 2 with a new one, but I can not see the difference between the old and new figures, and the latin names in the new figure is still not in italic. The way the authors used to represent the significant difference (a,b,c,a*,b*) is still confusing with no change, which makes their results obscure. For example, at line 499-501, the authors mentioned “In general, representatives of the Mytilidae Family differed from those of the Pectinidae Family in a lower level of ROS generation and a higher MDA content in the mitochondria of gill cells.” Is there statistical comparison between the two family in this figure? Moreover, how to explain that the ROS level is higher in long-lived species at constitutive level, but lower in long-lived species under oxidative stress?

   Moreover, for the comments 3 and 4, I can not find the exact response, and the authors may only delete the statement but without proper explanation. The bivalve species have one of the most significant diversity with different MLS and body size, and the age and body size of the representative species used in this study may have significant effect on the conclusion. From table 1, the body size, age, and age stage to MLS are all different. In Mytilidae, the age (24, 20, 4) and stage to MLS (24/150, 20/61, 4/6) are different among species; while in Pectinidae, the age is the same (5, 4, 4) but the stage to MLS (5/22, 4/15, 4/9) is different among species, how did the authors control the sample difference which could have significant effect on the results?

   All in all, the authors used limited materials (sample size and sample strategy, actually, more sampled species are recommended) with many variables but without strict control to test a complex theory, which makes the conclusion arbitrary. For example, the authors mentioned at line 241 “- to investigate whether the FAs of gill mitochondrial membranes affect the degree of their oxidative damage and MLS of species.” , which can not be explained by the current result. Therefore, I still recommend not to accept this manuscript.

Author Response

(The authors gave the same response as above.)

Round 3

Reviewer 3 Report

I appreciate that the authors put some efforts to response to my comments and to improve their manuscript. Even still with some flaws, the current version generally fullfills the requirment for publication in biology.